# Risk of Lead Exposure from Transport Stations to Human Health: A Case Study in the Highland Province of Vietnam

**DOI:** 10.3390/toxics7030048

**Published:** 2019-09-19

**Authors:** Quang Huu Le, Dung Duc Tran, Yi-Ching Chen, Huong Lan Nguyen

**Affiliations:** 1Department of Environmental Engineering, Da-Yeh University, Changhua 51591, Taiwan; 2Centre of Water Management and Climate Change, Vietnam National University, HoChiMinh 700000, Vietnam; 3Institute of Research and Development, Duy Tan University, Danang 550000, Vietnam

**Keywords:** heavy metals, soil, crops, bus station, lead

## Abstract

Lead (Pb) is a heavy metal that negatively affects human health. Many studies have shown the relationship between lead exposure and various human activities, of which automobile service stations with gasoline emissions are considered the main cause. However, a limited number of studies have specifically considered lead exposure from automobile stations in Vietnam, as well as its impact on human activities and the surrounding natural resources. The objective of this study was to assess the possible risks of lead exposure to the surrounding agricultural and non-agricultural farms of a bus station located in the center of Dalat city, Lamdong province, Vietnam. To address this objective, 45 samples were collected from the soil, irrigated water resources, and vegetable crops of areas both close to and far away from the bus station. These samples were tested using the atomic absorption spectrometry technique. Our findings demonstrated higher lead concentration levels from all three types of samples collected from areas near the bus station. Of which, soil and water samples showed higher than normal exposure values of lead, but these were still under the allowed limits established by the Vietnam standard. Different from the soil and water, vegetable samples surrounding the bus station presented greater lead contamination than the permitted limit. High risk quotient (RQ) indexes were detected to point out that accumulative consumption of leaded vegetables over time could cause lead poisoning and harm human health. This study not only provides significant inferential evidence of the risk of lead exposure to agricultural activities and human health in Vietnam, but also delivers a real-life example for a real-world context.

## 1. Introduction

Lead is a heavy metal that is toxic to humans. Lead exposure occurs through inhalation or by ingesting food or water. Lead damages multiple systems including the human nervous system. Children are especially vulnerable because lead crosses the blood–brain barrier and results in brain damage [1]. Lead poisoning results from inhaling or ingesting lead-based paint as well as from ingesting lead contaminated food or water [2]. Lead also causes long-term harm in adults, such as increased risk of miscarriage, stillbirth, premature birth, and low birth weight, as well as minor defects [3]. The effect of lead also permanently reduces children’s cognitive abilities, even when they are exposed to extremely low levels [4]. Notably, lead may cause oxidative stress, which has been considered a risk factor for diabetes [5,6].

In Vietnam, heavy metal pollution, especially lead exposure, occurs in many localities owing to a variety of different causes. These include metal lead mining [7], industrial wastewater from factories and industrial parks, municipal waste, abuse of pesticides and pesticide chemicals [8], production of batteries, and lead recycling activities. Lead is in the exhaust gas of gasoline engines, because tetraethyl lead is mixed into gasoline to avoid explosions, and in the areas near bus stations, where the concentration of cars is high, emissions from vehicles are very high.

In 2015, the lead concentration test results of the Institute of Occupational Health and Environment showed that samples of soil, water, air, and vegetables in Dong Mai mineral exploitation village were many times higher than the allowed standard [9]. Water samples collected at drainage channels and ditches exceeded the permitted level by 1000 times. Through testing by the National Center for Poison Control and Children’s Hospital, 378 children in Dong Mai village were found to be suffering from lead poisoning. In another study in Ho Chi Minh city [10], 311 children were tested to reveal that the average blood lead levels are 4.97 μg.dL^−1^ with 7% of the 311 testing samples are higher than 4.97 μg.dL^−1^. There is a correlation between lead content in soil, accumulation in crops, and consumer health [9,11].

Dalat is one of the major vegetable production areas of Vietnam, supplying vegetables and flowers to the markets of the southeastern region. Vegetable production reaches about 2 million tons per year, of which 60%–63% is consumed in the southeastern provinces [12]. In Dalat, there is a high possibility of lead accumulation in areas around bus stations, especially in the valley area. Lead exposure will be exacerbated in the long term owing to the ongoing use of gasoline, which is a source of lead emissions [13].

Lead may overflow or seep from bus stations or automobile service stations [14]. Soil is one of the principle and most important resources for providing quality food and quality environments for humans. Crops on urban contaminated soil, or in water or soil contaminated with municipal, domestic, or industrial waste, can absorb heavy metals in the form of mobile ions in the soil through the roots, or through absorption by the leaves. These metals are biologically absorbed in the roots, stems, fruits, seeds, and leaves of plants [15].

Our study is relevant to the above-mentioned problems. The present study aimed to assess the lead exposure of bare land and agricultural farms surrounding a bus station in Dalat city, Lamdong province. To do this, samples were collected from the various resources of soil, irrigation water, and crops from the two types of land areas, both nearby and far away from the bus station. In addition, samples of the fertilizers used by the farmers were also tested to clarify whether they were the main cause of lead exposure for the vegetable crops. This study sheds light on whether there is a potential impact of lead poisoning on human health from cultivating crops around an automobile service station, and on how to prevent lead poisoning from affecting vegetable crops in the first place.

## 2. Study Area

Dalat is the urban city of Lamdong province (Figure 1). It is located on Langbiang plateau, at a height of 1520 m above mean sea level. Although Dalat lies in the monsoon tropical zone, the climate has its own characteristics of the highlands, with a lowest average daily temperature of 15 °C and a highest of 24 °C. The two seasons consist of the rainy season from April to November and the dry season from December to March. It is sunny all year round. These climatic conditions allow Dalat to produce vegetables, flowers, and specialty fruits [16].

The total natural land area of Dalat City is 39,106 ha. The agricultural land in Dalat encompasses approximately 10,000 ha, which is divided into agricultural land of 5300 ha and intercropped land of 4678 ha. The agricultural production land is divided into two main groups. The Feralit group has low to medium fertility, but it contains easily digested phosphorus and some trace elements suitable for vegetables, flowers, and fruit trees. The other group is reddish-brown Feralit on basalts, which is suitable for cultivating root vegetables. The traffic conditions are relatively convenient for expansion of economic exchanges with key economic regions in the south, the central coast provinces, and the central highlands [16].

In 2015, Dalat’s population was 220,151 people, of which 90% of the population was living in urban areas and the remaining 10% was working in rural areas. The population density is 558 people·km^−2^. At present, Dalat agriculture attracts 38.5% of workers. Agricultural production in terms of the fields of cultivation is growing in area, increasing the number of crops, increasing in productivity, and increasing the quality of the agricultural products. Vegetable and flower growing are the strengths of the locality in particular and the country in general, with an average growth rate of 9.1% per year over the last five years. In Dalat, there are more than 20,000 production facilities and farmers participating in growing vegetables, including leafy vegetables (i.e., cabbage, lettuce, green cauliflower, celery, and spinach), root vegetables (i.e., potatoes, onions, and beetroot), and fruit vegetables (i.e., tomatoes, eggplant, and beans) [16].

For a tourist city, vehicles are crowded in Dalat, and the bus station area tends to be congested. Therefore, emissions from these vehicles being discharged into the residential areas around the bus station is inevitable. Moreover, the location of the bus station is a hilltop area where the land is cultivated by local people. The risk of lead contamination in the agricultural products produced there is increasing because of this cultivation. The use of leaded gasoline had been allowed by the Vietnamese government for a long time; therefore, awareness of this issue is crucial.

## 3. Materials and Methods

### 3.1. Collection of Samples

Two sampling sites were established in two areas (Figure 1). The first, site A, was located around 300–500 m from the bus station, and site B was situated a few kilometers away. Both sites were closed to vehicle access or any form of human activities that could generate waste.

Five sampling points at 50 m intervals were mapped to determine where to collect soil samples in the sampling area. A clean stainless steel tool was used to obtain soil from a depth of 0–30 cm. A control sample of soil from fallow land without cultivation was also taken.

At each of the five sampling points, soil samples were taken from five different positions and thoroughly mixed in a clean plastic bucket for representative sampling. From this, 0.5 kg was put into a plastic bag. The samples were then dried, crushed, and sieved with a 2 mm mesh, before being stored in labeled plastic bags until analysis. To identify the samples, S1A–S10A were labeled for ten soil samples collected at site A, whereas S1B–S10B were labeled for those ten taken from site B. Twenty samples were collected in total.

From each of ten soil sampling points of S6A–S10A and S6B–S10B, water and vegetable samples were collected. This means that ten water and vegetable samples each were taken. The 10 vegetable samples included cabbage, lettuce, and napa cabbage grown either in the vicinity of the bus station or far away from it, and were collected randomly using a stainless steel knife. The vegetables were then washed with distilled water and dried to a constant mass in an oven at a temperature of 700 °C. The dry samples were finally pulverized with agate stone and kept in plastic bags.

Ten irrigation water samples of 0.5 L each were collected directly from drilling wells and ponds. In both cases, water was taken directly from the tube connected to the wells or ponds, from which farmers drew directly for crop irrigation. Each sample was taken five times, and each time, 0.5 L was stored in a plastic container and then shaken well, before the final sample of 0.5 L was taken and kept in a plastic bottle.

Fertilizer samples were obtained from commonly used organic fertilizers. Samples were taken from different bags, in accordance with the Vietnamese criterion of TCVN 9486:2013, and mixed well to obtain an average sample before being placed in plastic bags.

### 3.2. Sample Treatment

#### 3.2.1. Soil Sample

The soil samples required the following treatment steps. First, 5.0 g of dried soil and crushed sample were placed in a 100 mL volumetric flask. A total of 20 mL of extraction solution was added (0.05 N HCl + 0.025 N H_2_SO_4_). The flask was then placed in a mechanical shaker for 15 min. After that, it was filtered through filter paper with a 50 mL volumetric flask, before being diluted with the extraction solution [17].

#### 3.2.2. Vegetable Samples

The samples were weighed; the mass of the dry sample was determined; and then the samples were finely ground and put into polyethylene vials, covered, labeled, and stored in a dry and well-ventilated place. After preliminary treatment, 0.5 g of the the inorganic samples was weighed and then put in a flask, and 15 mL of concentrated HNO_3_ solution and 5 mL of solution condensed HCl were added. Each bottle was covered with watch glasses and left overnight. Then, the samples were cooked on a stove a small distance from the sand until the solution was clear and there was no reddish-brown gas escaping. The samples were transferred to glass beakers to continue boiling to dry and expel the excess acid. HNO_3_ solution was used to dissolve the samples, and they were transferred quantitatively to 10 mL volumetric flasks.

### 3.3. Analytical Process

The digested samples were analyzed for lead using an atomic absorption spectrophotometer (AAS VGB 210 System) in the Center for Analysis Service of Experiment in Hochiminh city, Vietnam. The instrument settings and operational conditions were set in accordance with the manufacturer’s specifications.

### 3.4. Risk Quotient Assessment

A risk quotient (RQ) is the ratio of a point estimate of exposure and a point estimate of effects. If a RQ value is less than 1, then there is an acceptable risk. In contrast, if the RQ is greater than 1, there is an unacceptable level of risk, which needs measures to reduce exposure. It is calculated to assess human health risk, using the following equation:(1)$$R=\frac{PEC}{PNEC},$$
where PEC is the measured environmental concentration;

PNEC is the limit concentration or the predicted no effect concentration [18]. In this study, the Vietnamese have used a standard limit of 0.3 mg.kg^−1^ for vegetables [19].

## 4. Results

### 4.1. Risk of Lead Exposure from Soil

Figure 2 presents the lead exposure in mg·kg^−1^, analyzed from sampling points on bare land and vegetable cultivation areas both surrounding and far away from the bus station. The horizontal line at 70 mg·kg^−1^ is the permissible limit of lead in soil.

In general, the lead concentrations were found to be two to three times higher from samples collected from bare land near the bus station than those from the far away bare land. Specifically, the lead concentrations in soil from bare land near the bus station ranged from 20 to 68 mg·kg^−1^, whereas the lead concentrations ranged from 0.2 to 21 mg·kg^−1^ in soil collected from bare land far away from the bus station. Regarding vegetable cultivation land, the lead exposure in the soil fluctuated in both the nearby and far away land areas, but their quantities ranged from 18 to 33 mg·kg^−1^. Despite the given differences, the lead concentrations in the soil from all collected samples were still under the permissible limit.

### 4.2. Risk of Lead Exposure from Irrigated Water and Vegetables

Higher lead concentrations were found in the water samples collected from water resources, such as springs, ponds, and drilling wells, in the areas surrounding the bus station than areas further away (Figure 3). The concentrations varied in the range of 0.25 to 0.8 µg·L^−1^ in the water samples near the bus station, but fluctuated in a range of 0.2 to 0.42 µg·L^−1^ for those far away from the bus station. Although higher concentrations were identified from irrigated water resources in areas near the bus station, these were still much smaller than the lead limit permissible for irrigated water (50 µg·L^−1^).

Regarding the lead concentrations measured from the vegetable samples, those collected from cultivation land near the bus station were threefold higher than those far away from the bus station (Figure 4). The lead concentrations in vegetables changed from 0.3 to 0.7 mg·kg^−1^ and from 0.03 to 0.3 mg·kg^−1^ in the areas near and far away from the bus station, respectively. Different with the soil and water samples, the lead concentrations found in the vegetables were higher than the permissible limit of 0.3 mg·kg^−1^.

Table 1 shows risk quotient indexes (RQ) calculated for vegetables samples. The values indicate the higher risk from lead exposure of vegetables grown in bus stations than those in remote areas. The RQ increases proportionally to the distance of close to far from the bus stop to the sampling points. High risk found from the lands where vegetables have been grown near the bus station. The results of analysis point out the level of risk to consumers’ health. Fortunately, the surveyed locations surround the Dalat bus station at present have converted from crops to flowers. Another reason informed by local people was that they have no longer cultivated vegetables owing to the urbanization affected.

### 4.3. A Test for Checking Potential Lead Exposure from Fertilizer

Table 2 is presented in order to clarify the contribution of fertilizer to lead exposure to soil and vegetables. As shown in the table, five samples were taken from various fertilizers used by farmers. The lead concentrations were within the range of 9.2 to 26.1 mg·kg^−1^, which is under the permitted limit of 200 mg·kg^−1^. According to the farmers, these fertilizers were all registered with responsible agencies.

Raw data are present in the Appendix A.

## 5. Discussion

### 5.1. Key Findings

Our findings reveal higher concentrations of lead from the samples collected at three main sources in an area close to the bus station. First, the three times higher concentrations of lead from bare-land soil samples around the bus station indicate the possibility of lead effects owing to the use of leaded gasolines. Second, greater lead concentrations were also found in the samples of irrigated water collected from ponds, drilling wells, and rainwater springs near the bus station. Third, the vegetable samples showed the highest levels of lead from the areas near to compared with those far away from the bus station. These findings indicate that the lead exposure may occur as a result of lead-contaminated rainwater affected by gasoline waste running from the top-hill station source [20].

The research findings, moreover, divulge a close relationship between the risk of lead exposure and the distance of the sampling point to the bus station. In other words, the closer the sample collection was to the bus station, the higher the lead concentrations in the soil, water, and vegetable samples. To conclude, the findings from samples collected from the three main sources imply a reasonably causal relationship between the risk of lead exposure to human activities and leaded gasoline waste from automobile stations, as shown by previous studies [14,21,22].

Lead absorption by vegetables should also be taken into account because it is considered as beneficial contribution to the human food supply [23]. The findings represented no difference in lead exposure between soil samples collected from the vegetable-growing lands surrounding and far away from the bus station (Figure 2). This raises the possibility that the lead is absorbed by vegetable bodies, reducing levels in the soil. This is supported by looking at the higher lead concentrations in the vegetable samples from the cultivation lands surrounding the bus station compared with those collected from further land (Figure 4).

Even though the lead concentrations in the soil and water samples in this study were mostly under their permissible limits, people living in the area adjacent to the bus station should be aware of the potential long-term effects on their health owing to the higher than limit lead exposure found from vegetable samples. The continuous usage of the same farms for growing crops will lead to bioaccumulation of lead over time, which has been associated with health risks [22]. Although it came up recently that the vegetable area nearby the bus station investigated in this study does not exist anymore, the risk is still there for people living around. If the levels of lead in the soil and water increase over time or even spread to further areas, it may become difficult to manage the safety of local residents.

### 5.2. Implications of the Findings in a Wider Context

This study explored the risk of lead poisoning based on in situ data collected from various sources of soil, irrigated water, and vegetables around a bus station. Many studies have publicized the impacts of lead poisoning from leaded gasolines on health, especially the harm to pregnant women and children [24,25,26,27]. Lessons learnt from the present study were gained by practical, but effective approaches for detecting lead exposure. The findings will be useful for people in the area, especially those who are living near the sources of lead contamination. These people have been drinking water and consuming agricultural products, which may be affected by lead exposure from the usage of leaded soil and water during the growing periods. Moreover, the local governments should use these findings to justify more attention being paid to the impacts of industrial activities near residential areas.

Several studies have recognized the risk of leaded petrol from automobile service stations or from motorways, although their contexts of study and method designs were different [14,21]. Our study found higher lead levels than normal in the soil, water, and vegetables in the bus station areas, which was also highlighted by Hashim et al. [21]. In their study, automobile exhausts were specified as the main source of air pollution containing heavy metals, particularly lead, in areas surrounding the highway. This was the result of adding organic lead compounds such as tetraethyl lead and tetramethyl lead to petrol, which also occurs in our study area. In other research, Teju et al. [14] conveyed the levels of lead pollution in roadside soils as lead concentrations of 13.5 ± 0.3, 19.6 ± 0.7, and 20.8 ± 0.8 µg·g^−1^ in three towns within one kilometer of the main roads. Hence, our findings of 20 to 68 mg·kg^−1^, which is equivalent to 20 to 68 µg·g^−1^, reflect the reasonably significant effects of lead exposure to land close to the bus station area.

The study’s findings also raise awareness of people if the lead concentrations are accessed relative to the internationally permissible limits. The mean Pb concentrations reported in agricultural soils in the USA [28], Europe [29], Poland [30], and Iran [31] are 12.3 mg.kg^-1^, 16 mg.kg^-1^, 16.4 mg.kg^-1^, and 6.1 mg.kg^-1^, respectively. According to these standards, the levels of Pb contamination in agricultural soil in our study area are comparatively higher.

### 5.3. Research Limitations and Future Research

This study has a limited number of samples owing to a limited budget. However, it collected sampling data from various principle resources, revealing valuable evidence of lead exposure. Additionally, the study indicated a simple, but practical method to identify the causes of lead exposure from a bus station. Future studies could collect more data to replicate our research findings. In addition, other types of contamination than lead metal should be investigated, in order to comprehensively assess the negative impacts on soil, water, and vegetables from lead exposure from the bus station.

Further investigations are, however, required in terms of human health testing. Specifically, data needed to be investigated on a number of lead-affected people in lead contaminated areas, or even further data from the local people living in the areas surrounding the bus station where potentially harmful lead levels are found. This would add weight to our findings. Moreover, similar studies could be undertaken with the same methodology for other areas where a similar bus station or other source of contamination exists.

## 6. Conclusions

This study presented the findings from data collected from multiple sources in the area surrounding a bus station. The findings highlighted increased lead levels in areas near the bus station. Specifically, the sampling of water, soil, and vegetables exposed higher lead concentrations in areas closer to the bus station. Lead exposure will increase in the long term from sources such as this, potentially to a level that has a negative influence on human health, especially for pregnant women and children. The authors make the following conclusions:In the short term, the lead poisoning level may be too low to be harmful to human health and safety, but its impact will be visible in the long term.Human activities, especially agricultural crop production, should be undertaken far away from industrial areas such as bus stations. Otherwise, monitoring systems should be installed in the cultivation areas to warn farmers should the lead concentration exceed the permissible limit. Such systems could detect an issue in time to prevent dangerous lead exposure.Land use planning needs to be implemented under the strict regulation of authorities and the national government to protect residential areas, especially areas near industrial zones that generate lead contamination.

## Figures and Tables

**Figure 1 toxics-07-00048-f001:**
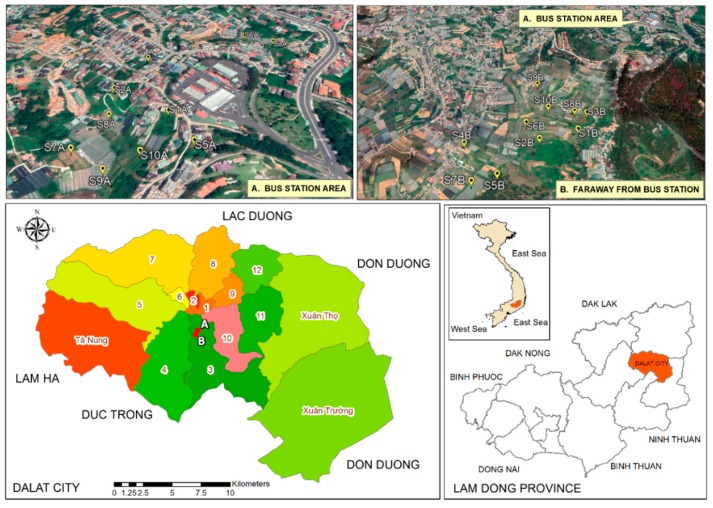
Study area in Dalat city and the location of sampling points.

**Figure 2 toxics-07-00048-f002:**
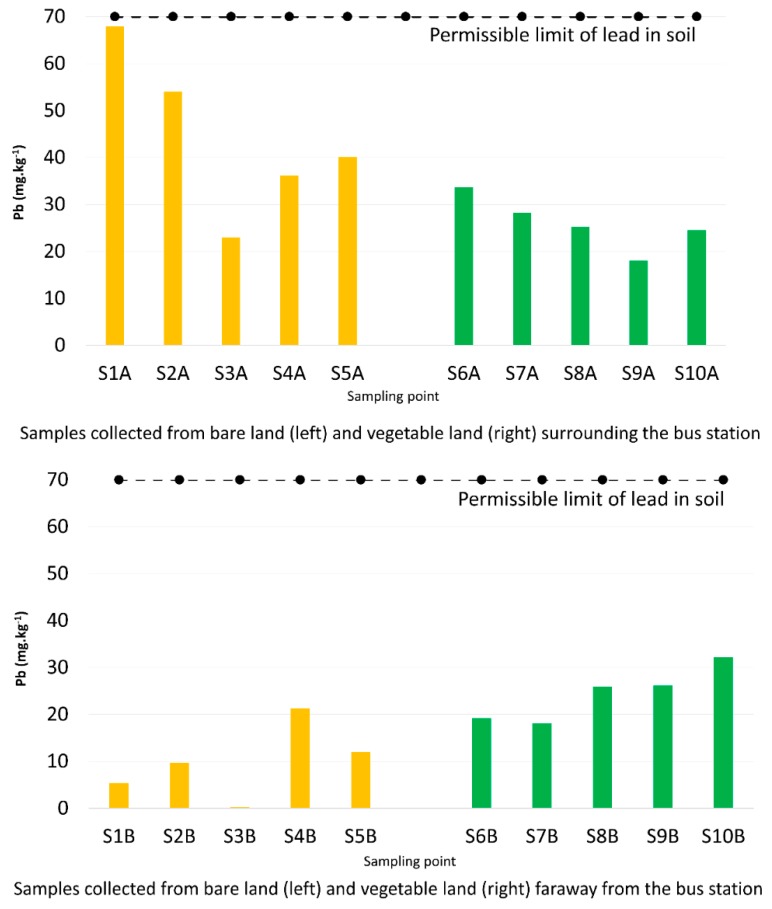
Comparison of soil lead concentrations measured from the two types of land areas near to and far away from the bus station.

**Figure 3 toxics-07-00048-f003:**
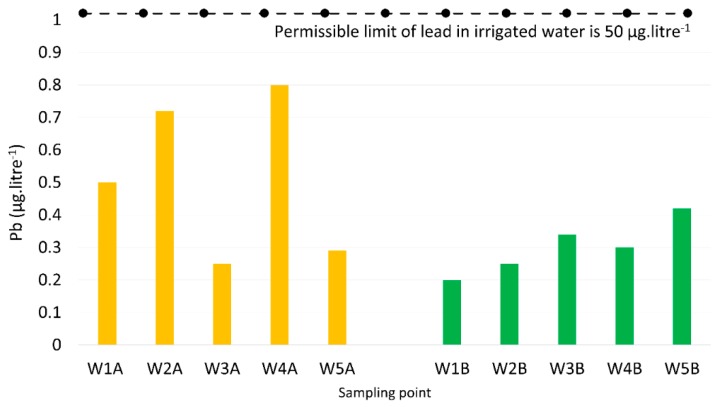
Lead concentrations of samples measured from irrigated water resources near (**left**) and far away (**right**) from the bus station.

**Figure 4 toxics-07-00048-f004:**
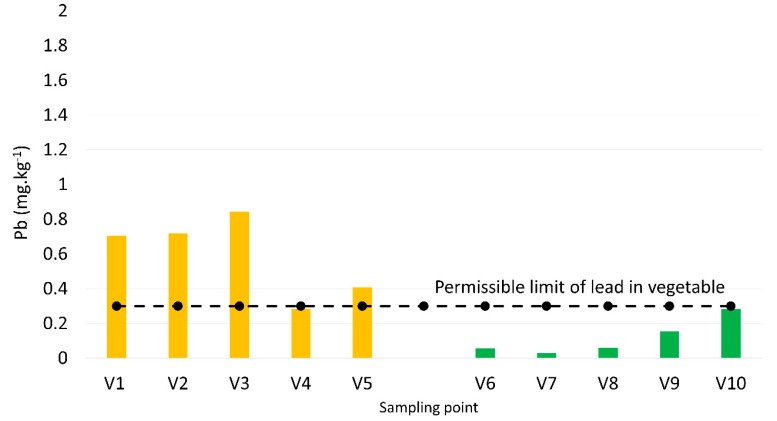
Lead concentrations of samples from vegetables planted in land near (**left**) and far away (**right**) from the bus station.

**Table 1 toxics-07-00048-t001:** Risk quotient (RQ) indexes calculated from vegetable samples.

Sample	RQ	Risk Level Index		
		Very Low Risk 0.01–0.1	Medium Risk 0.1–1	High Risk ≥1
**Near the bus station area**				
V1	2.35			x
V2	2.40			x
V3	2.82			x
V4	0.94		x	
V5	1.36			x
Far away from the bus station area				
V6	0.18		x	
V7	0.10		x	
V8	0.20		x	
V9	0.51		x	
V10	0.94		x	

**Table 2 toxics-07-00048-t002:** Fertilizer samples used by farmers.

Samples of Fertilizer	P1	P2	P3	P4	P5
Registration component	OM: 15%; N: 6%; P_2_O_5_: 3%; K_2_O: 3%; Humidity: 25%; pH_H2O_: 5.	OM: 65%; N: 3.1%; P_2_O_5_: 1.8%; K_2_O: 1.8%; Humic acid: 3%; Mg: 0.3%; Ca: 3%; Zn, Fe, Mn, Cu, B were barely detected.	OM: 22.4%; N: 4%; P_2_O_5_: 3%; K_2_O: 2%; Humic acid: ≥3.5%.	OM: 22.4%; N: 3%; P_2_O_5_: 3%; K_2_O: 3%; Humic acid ≥3.5%.	Fresh fish No registration and recommended not to use
Limit heavy metal registration	As ≤ 10 mg·kg^−1^; Cd ≤ 5 mg·kg^−1^; Pb ≤ 200 mg·kg^−1^; Hg ≤ 2 mg·kg^−1^				
Analyzed results of Pb content (mg·kg^−1^)	9.21	10.53	2.32	26.12	23.65

OM: Organic Matter, Permissible limit accepted by the Vietnamese Standard is 200 (mg·kg^−1^).

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
