# Peer review of "Risk of Lead Exposure from Transport Stations to Human Health: A Case Study in the Highland Province of Vietnam"

_toxics, 2019, doi:10.3390/toxics7030048_

Round 1
Reviewer 1 Report
Lines 36-37: "When breathed in, eaten, or drinking lead toxicity
37 can harm the human nervous system, especially in children, and cause blood and brain disorders." should be replaced by "Lead exposure occurs through inhalation or by injesting food or water. Lead damages multiple systems including the human nervous system. Children are especially vulnerable because lead crosses the blood-brain barrier and results in brain damage."
Line 38" "Lead poisoning mainly results from either leaded food or drinking water [1]" should be changed to "Lead poisoning results from inhaling or injesting lead-based paint as well as from injesting lead contaminated food or water."
Line 54 "be suffering lead poisoning" should be changed to "be suffering from lead poisoning"
Line 66 "Lead may overflow or seep from bus stations or automobile service stations." Other studies have demonstrated the lead is released in the emissions from the vehicles and distributed over adjacent soil. If the authors agree, they should add a citation about that. There are many studies about soil lead close to high traffic areas.
Can the map also show the water sampling and vegetable sampling locations?
Can the authors compare the Vietnamese lead standards with other standards internationally, given that this is an international audience? For example, soil lead levels for children's play areas are 0.4 mg/kg which is significantly lower in US. It might make more sense to compare to other countries standards.
Can the authors include a table that shows all the data together and includes descriptive statistics?
Can the authors include some information in the background about children's blood lead levels in Vietnam and if possible in the particular province?
Author Response
Point 1: Lines 36-37: "When breathed in, eaten, or drinking lead toxicity can harm the human nervous system, especially in children, and cause blood and brain disorders." should be replaced by "Lead exposure occurs through inhalation or by ingesting food or water. Lead damages multiple systems including the human nervous system. Children are especially vulnerable because lead crosses the blood-brain barrier and results in brain damage."
Response 1: The authors thank the reviewer for the valuable comments. In the revised manuscript, we have replaced the sentence you recommended.
Point 2: Line 38" "Lead poisoning mainly results from either leaded food or drinking water [1]” should be changed to "Lead poisoning results from inhaling or ingesting lead-based paint as well as from ingesting lead contaminated food or water."
Response 2: Please find the authors’ change based on the recommendation.
Point 3: Line 54 "be suffering lead poisoning" should be changed to "be suffering from lead poisoning"
Response 3: The phrase has been replaced in the revised manuscript.
Point 4: Line 66 "Lead may overflow or seep from bus stations or automobile service stations." Other studies have demonstrated the lead is released in the emissions from the vehicles and distributed over adjacent soil. If the authors agree, they should add a citation about that. There are many studies about soil lead close to high traffic areas.
Response 4: Many thanks. The authors totally agree with the reviewer on this point. A citation therefore has been added.
Point 5: Can the map also show the water sampling and vegetable sampling locations?
Response 5: Thank for your comment. Actually, the 10 water sampling locations and 10 vegetable sampling locations were at the same locations where 10 of 20 soil samples were collected. From the map, these water and vegetable sampling locations are at S6A-S10A and S6B-S10B. Therefore, the authors have clarified these in Section 3 (Materials and Methods) in the revised manuscript, by adding text from the lines 119-123. In Figure 1, the authors changed “location of soil sampling points” from the caption to be “location of sampling points”.
Point 6: Can the authors compare the Vietnamese lead standards with other standards internationally, given that this is an international audience? For example, soil lead levels for children's play areas are 0.4 mg/kg which is significantly lower in US. It might make more sense to compare to other countries standards.
Response 6: The authors have added text to the revised manuscript (Discussion section, lines 269-273).
Please find the addition from the lines 269-273.
Point 7: Can the authors include a table that shows all the data together and includes descriptive statistics?
Response 7: The authors have included all data with descriptive statistics in Tables A1 and A2 attached to the Appendix of the manuscript.
Point 8: Can the authors include some information in the background about children's blood lead levels in Vietnam and if possible in the particular province?
Response 8: Thanks the reviewer for this comment. We have added necessary text in the Introduction section to clarify (lines 54-56 were added).
Reviewer 2 Report
Dear Authors
In my opinion the theme of the article is very actual and interesting for the readers of the journal.
The authors assessed the possible risks of lead exposure to the surrounding agricultural and non-agricultural farms of a bus station located in the center of Dalat city, of Lamdong province, Vietnam.
Forty five samples were collected from the soil, irrigated water resources, and vegetable crops of areas both close to and far away from the bus station.
The authors found that the sampling of water, soil, and vegetables showed higher lead concentrations in the areas closer to the bus station.
This research demonstrated high lead concentration levels from all three-type samples, particularly in vegetable samples (above the Vietnam admitted limits), collected from areas near the bus station.
High risk quotient indexes were found by long term Pb exposure, mainly by consumption of vegetables, which over time could cause lead poisoning and harm human health.
The authors concluded that human activities, especially agricultural crop production, should be undertaken far away from industrial areas such as bus stations. Otherwise, monitoring systems should be installed in the cultivation areas to warn farmers should the lead concentration exceed the permissible limit. Such systems could detect an issue in time to prevent dangerous lead exposure.
This study show that land use planning needs to be implemented under the strict regulation of authorities and the national government to protect residential areas, especially areas near industrial zones which generate lead contamination.
Please added in L38 the reference:
Cabral-Pinto, M. M. S., Ordens, C. M., de Melo, M. T. C., Inácio, M., Almeida, A., Pinto, E., & da Silva, E. A. F. (2019). An Inter-disciplinary Approach to Evaluate Human Health Risks Due to Long-Term Exposure to Contaminated Groundwater Near a Chemical Complex. Exposure and Health, 1-16.
The paper is well structured, the title and abstract clearly describe the content of the manuscript, and the language is correct and clear. However, in my opinion, after Discussion, it is necessary a new Section with the conclusions. Also thee Tables must be formatted, according with the Journal Instructions.
In my opinion minor revision is needed.
Best regards
Author Response
Point 1: Dear Authors
In my opinion the theme of the article is very actual and interesting for the readers of the journal.
The authors assessed the possible risks of lead exposure to the surrounding agricultural and non-agricultural farms of a bus station located in the center of Dalat city, of Lamdong province, Vietnam.
Forty-five samples were collected from the soil, irrigated water resources, and vegetable crops of areas both close to and far away from the bus station.
The authors found that the sampling of water, soil, and vegetables showed higher lead concentrations in the areas closer to the bus station.
This research demonstrated high lead concentration levels from all three-type samples, particularly in vegetable samples (above the Vietnam admitted limits), collected from areas near the bus station.
High risk quotient indexes were found by long term Pb exposure, mainly by consumption of vegetables, which over time could cause lead poisoning and harm human health.
The authors concluded that human activities, especially agricultural crop production, should be undertaken far away from industrial areas such as bus stations. Otherwise, monitoring systems should be installed in the cultivation areas to warn farmers should the lead concentration exceed the permissible limit. Such systems could detect an issue in time to prevent dangerous lead exposure.
This study shows that land use planning needs to be implemented under the strict regulation of authorities and the national government to protect residential areas, especially areas near industrial zones which generate lead contamination.
Response 1: The valuable comments of the reviewer are highly appreciated.
Point 2: Please added in L38 the reference:
Cabral-Pinto, M. M. S., Ordens, C. M., de Melo, M. T. C., Inácio, M., Almeida, A., Pinto, E., & da Silva, E. A. F. (2019). An Inter-disciplinary Approach to Evaluate Human Health Risks Due to Long-Term Exposure to Contaminated Groundwater Near a Chemical Complex. Exposure and Health, 1-16.
Response 2: Based on the reviewer’s comment, the citation has been added to the L38 of the revised manuscript (the citation no.1 from the reference list).
Point 3: The paper is well structured, the title and abstract clearly describe the content of the manuscript, and the language is correct and clear. However, in my opinion, after Discussion, it is necessary a new Section with the conclusions. Also thee Tables must be formatted, according with the Journal Instructions.
Response 3: Thanks for your positive comments. Actually, the Discussion in the original manuscript was separated with the Conclusions. Your recommendation on formatting the Tables has been considered following the Journal Instructions.
Round 2
Reviewer 1 Report
Can be accepted